# A pumpless liver-adipose model for studying metabolic dysfunction and drug responses

Zeinab Ebrahimian[1,2], Fatemeh Kalalinia[3,4], Amir Reza Ameri[2,5], Hossein Hosseinzadeh[1,6]*, Bibi Marjan Razavi[1,7], Seyed Ali Mousavi Shaegh [ID][2,8,9,10]*

1 Department of Pharmacodynamics and Toxicology, School of Pharmacy, Mashhad University of Medical Sciences, Mashhad, Iran, 2 Laboratory for Microfluidics and Medical Microsystems, Research Institute for Medical Sciences, Mashhad University of Medical Sciences, Mashhad, Iran, 3 Biotechnology Research Center, Pharmaceutical Technology Institute, Mashhad University of Medical Sciences, Mashhad, Iran, 4 Department of Pharmaceutical Biotechnology, School of Pharmacy, Mashhad University of Medical Sciences, Mashhad, Iran, 5 School of Biomedical Engineering, University of British Columbia, Vancouver, Canada, 6 Pharmaceutical Research Center, Department of Pharmacodynamics and Toxicology, School of Pharmacy, Mashhad University of Medical Sciences, Mashhad, Iran, 7 Targeted Drug Delivery Research Center, Department of Pharmacodynamics and Toxicology, School of Pharmacy, Mashhad University of Medical Sciences, Mashhad, Iran, 8 Department of Biomedical Engineering, Mashhad University of Medical Sciences, Mashhad, Iran, 9 Clinical Research Unit, Ghaem Hospital, Mashhad University of Medical Sciences, Mashhad, Iran, 10 Orthopedic Research Center, Mashhad University of Medical Sciences, Mashhad, Iran

* hosseinzadehh@mums.ac.ir (HH); Mousavisha@mums.ac.ir (SAMS)

## Abstract

Metabolic dysfunction is a primary driver of chronic diseases, such as obesity and type 2 diabetes. Developing *in vitro* models allows to understand metabolic mechanisms and develop effective treatments. To this end, a pumpless open-top co-culture (POCC) device to simulate liver–adipose tissue interactions was developed. The POCC comprises a four-chamber bioreactor with an open-top configuration and an easy-to-assemble cover cap. Rotational motion enabled perfusion of culture media among the interconnected chambers. HepG2 and 3T3-L1 cells were co-cultured and exposed to olanzapine (Olz), chlorogenic acid (CGA), and metformin (Met) for four days. Drug responses in the POCC were compared with those in 96-well mono-cultures. Assessments were conducted for glucose/triglyceride (TG) levels, lipid accumulation, and cell viability. While viability remained unchanged, Olz increased lipid content, extracellular glucose and TG levels in the POCC model, effects that CGA and Met mitigated. This practical and physiologically relevant platform offers a promising alternative for preclinical screenings that could be employed to study other multi-tissue interactions in various disease models.

## 1. Introduction

Metabolic dysfunction underlies many chronic diseases, including obesity, type 2 diabetes, and non-alcoholic fatty liver disease, a significant burden on global health [1,2]. These disorders arise from the disruption of intricate and tightly regulated

**Data availability statement:** All relevant data are within the paper and its Supporting Information files.

**Funding:** The authors are grateful to the Vice Chancellor of Research, Mashhad University of Medical Sciences (No. 971985), Mashhad, Iran for financial support. The funders had no role in study design, data collection and analysis, decision to publish, or preparation of the manuscript.

**Competing interests:** The authors have declared that no competing interests exist.

metabolic networks that involve crosstalk among the liver, adipose tissue, muscle, and pancreas [3]. Multiple factors, such as an unhealthy lifestyle, genetic predisposition, and physical inactivity, contribute to metabolic dysfunction [4,5]. Drug-induced metabolic disturbances have been a significant factor. Olanzapine (Olz), a second-generation antipsychotic, has been shown to cause weight gain, insulin resistance, and dyslipidemia [4,6]. To manage the occurrence of these complications by reducing hepatic glucose production, metformin (Met), an antidiabetic agent, is usually co-prescribed to counteract these effects in patients taking this drug [7–9]. Preclinical studies have shown the anti-lipogenic and anti-diabetic potential of natural compounds, such as chlorogenic acid (CGA) [10,11]. Predicting the effects of drugs on metabolic pathways remains challenging owing to the limitations of preclinical models [12]. Animal models frequently fall short of predicting human drug metabolism because of pharmacokinetic and pharmacodynamic differences between humans and animals [13]. The current *in vitro* models lack the inter-organ dynamics and physiological complexity [14], or their complexities have made the manufacturing challenging [15].

The bioreactor platforms have the ability to replace *in vivo* models for mechanistic studies [16]. To enhance physiological relevance and high-content drug screening, they can incorporate various cell types into networked microscale environments with dynamic perfusion [17–23]. Scalability, reproducibility, and cost-effectiveness are some of the advantages of these devices [24]. Modeling metabolic diseases has been made possible using multi-bioreactor platforms that incorporate metabolically relevant tissues, such as the liver, adipose tissue, and pancreas [25–29]. For instance, liver-gut systems were created to model the gut–liver axis in non-alcoholic fatty liver disease [30,31] and pancreas–liver axis in diabetes [28,32]. Metabolic crosstalk in muscle–adipose, liver-adipose, and immune-adipose cells has also been studied in multi-bioreactor systems [33]. These miniaturized and enclosed design systems depend on external or integrated micropumps, such as peristaltic [18,34] or on-board pumps [35], to circulate the culture medium. The use of micropumps increases the risk of air bubble formation, clogging, and operational accessibility in conventional incubators [18,34–38]. This dependence on an external pump adds technical complexity to the enclosed and compact designs of microfluidic platforms [13,39].

In this work, we developed a pumpless open-top co-culture (POCC) system that models liver–adipose metabolic interactions. This platform eliminates the need for external pumps by utilizing a biochip with connected open-top bioreactors positioned on a specially designed rotator to enable passive medium circulation through rotational motion. This setup facilitates simple medium exchange, imaging, sampling, and cell seeding. We used this model to study the metabolic responses of a co-culture of 3T3-L1 adipocytes and HepG2 hepatocytes to Olz, CGA, and Met. We evaluated the effects of drugs on cell viability, intracellular lipid accumulation, and extracellular levels of glucose and triglycerides (TG). The POCC model successfully reproduced the expected metabolic outcomes and showed good agreement with findings from our previous *in vivo* studies.

## 2. Materials and methods

### 2.1. Materials

Poly (methyl methacrylate) (PMMA) sheets (Cho Chen, Taiwan) and thermoplastic polyurethane (TPU; Dureflex PT9200, Covestro, Leverkusen, Germany) were used as primary materials. Phosphate-buffered saline tablets (PBS) and resazurin (AlamarBlue®) were purchased from Sigma-Aldrich (USA). Trypsin, penicillin/streptomycin, and Dulbecco's modified Eagle medium (DMEM; high glucose) were obtained from Bio-Idea (Iran). Fetal bovine serum (FBS) was purchased from Gibco (Darmstadt, Germany). 3T3-L1 cells (mouse embryonic fibroblasts) were acquired from the Cell Resource Center at Ferdowsi University of Mashhad (Mashhad, Iran), whereas human hepatocellular carcinoma cell lines (HepG2 cells) were obtained from the School of Pharmacy, Mashhad University of Medical Sciences. Olanzapine (Olz) was obtained from Sobhan Co., Iran, and chlorogenic acid (CGA) was obtained from Tinab Shimi Co., Iran. Metformin (Met) was obtained from MAHBAN Chemi Co. (Tehran, Iran). The oil red o (ORO) staining lipid kit was prepared from BioVision (USA).

### 2.2. Chip design

A whole-thermoplastic co-culture device was made using PMMA sheets that enabled the simultaneous cultivation of multiple cell lines in four interconnected chambers with controllable circulation of the culture medium. The co-culture device consisted of a multichamber bioreactor chip and its covering cap (Fig 1A). The bioreactor chip was fabricated by assembling five layers of PMMA sheets to create four open-top cell culture chambers (Fig 1B). These chambers had an oval geometry with diameters of 5 and 10 mm and a depth of 5.5 mm (Fig 1C). The open top (i.e., no top cover) design of the chambers with interconnected channels facilitates direct cell seeding in bioreactors as well as gas exchange between culture medium and the surrounding environment. In addition, the external rotator enables intermittent or continuous recirculation of the culture medium through the biochip. The elliptical bioreactor geometry was designed to promote the uniform movement of the culture medium while minimizing bubble entrapment. Each bioreactor had a surface area of 39.236 mm² and a volume of approximately 200 µL, which was comparable to the wells of a 96-well plate. The bioreactor chip was developed to reduce shear stress and cell detachment while facilitating the exchange of cell-secreted factors between chambers through regulated rotational movement. The interconnecting channels had a rectangular cross-section measuring 2 mm width and 4 mm in deep, which was optimized to allow efficient fluid displacement during rotation without generating excessive mechanical forces.

To prevent evaporation and potential contamination, the bioreactor chip was covered with a cap designed as a five-layer assembly incorporating a TPU polymer film and PMMA, as illustrated in Fig 1D. TPU is a gas-permeable material that allows and maintains efficient gas exchange and optimal conditions for cell growth.

For periodic fluid displacement within the culture chambers, an in-house rotator with an adjustable rotation speed and direction was fabricated (Fig 2A-2B). To circulate the culture media, the bioreactors were mounted on the upper plate of the rotator, and the entire system, including the rotator and bioreactors, was placed inside the incubator (Fig 2C). The rotator was enclosed in a sealed casing to withstand the high humidity of the incubator, and its control unit was designed to be externally positioned to allow easy operation.

### 2.3. Chip fabrication

The bioreactor chips and caps were made of PMMA sheets of various thicknesses and TPU films with a thickness of 25 µm using our previously developed method [40,41]. The chips were designed by SOLIDWORKS® software (Version 2017), and PMMA sheets were cut by a laser machining (Almas Sanat Pishro Khorasan Co, Iran). TPU films were cut to the desired dimensions using a sharp blade. The components were cleaned, washed, and preheated for ~ 8 h at 90 °C in a vacuum with a gauge pressure of −0.8 kPa. They were then assembled and bonded via low-pressure thermal-fusion bonding at 145 °C for ~ 2 h in a vacuum with a gauge pressure of −0.8 kPa. The preheating and thermal fusion bonding

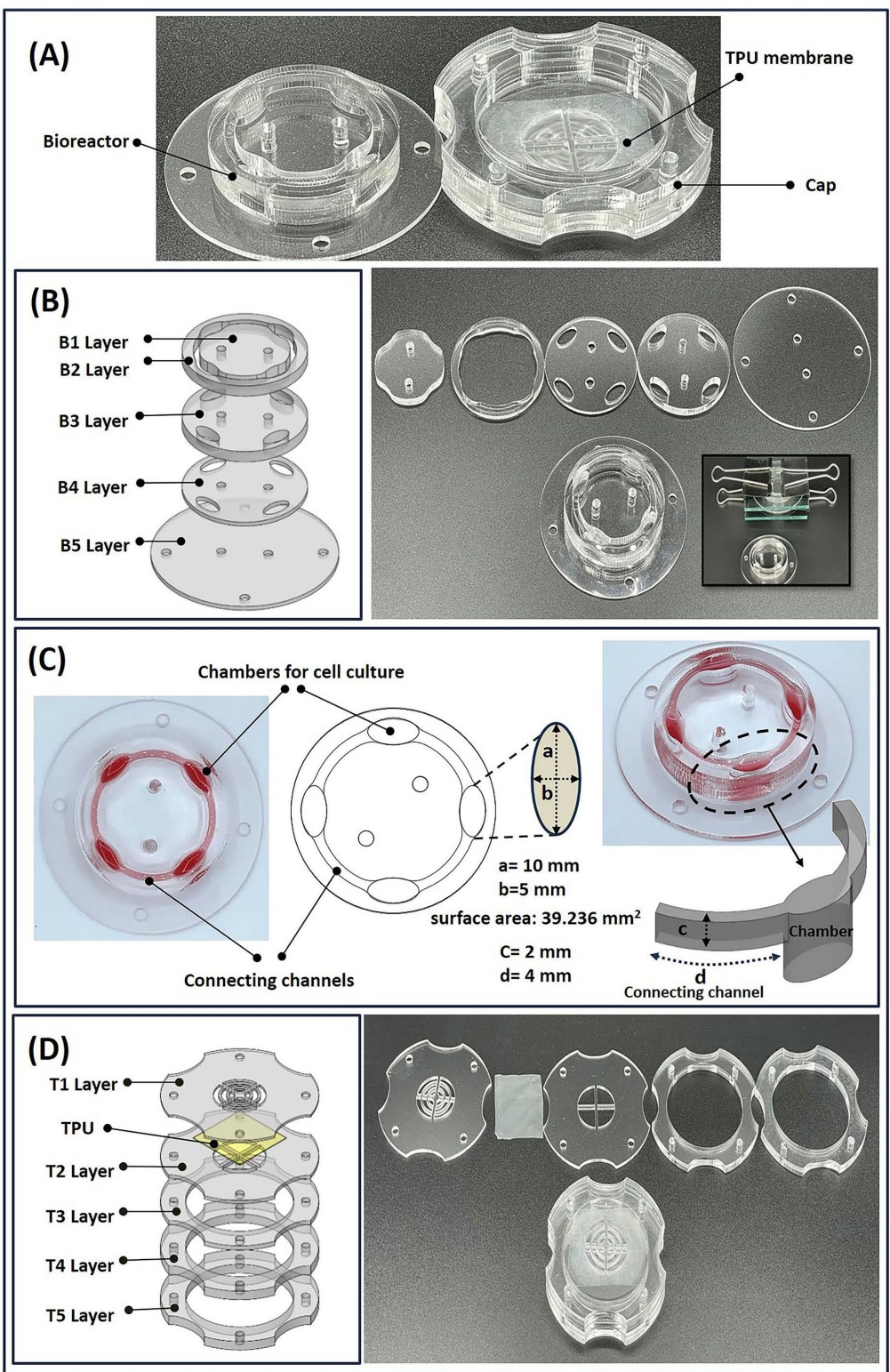

**Fig1. The co-culture device (POCC) design and assembly. (A)** POCC chip consists of a bioreactor and a covering cap. **(B)** The schematics of five layers of the bioreactor part (left), along with the laser-cut of the bioreactor components and its assembly (right). **(C)** The top view of the bioreactor with four oval chambers (10 mm × 5 mm, 39.236 mm² surface area) connected by rectangular channels (2 mm wide, 4 mm deep). **(D)** The schematics of the five layers and the TPU (thermoplastic polyurethane) of the cap part (left), as well as the laser-cut of the cap components and its assembly (right).

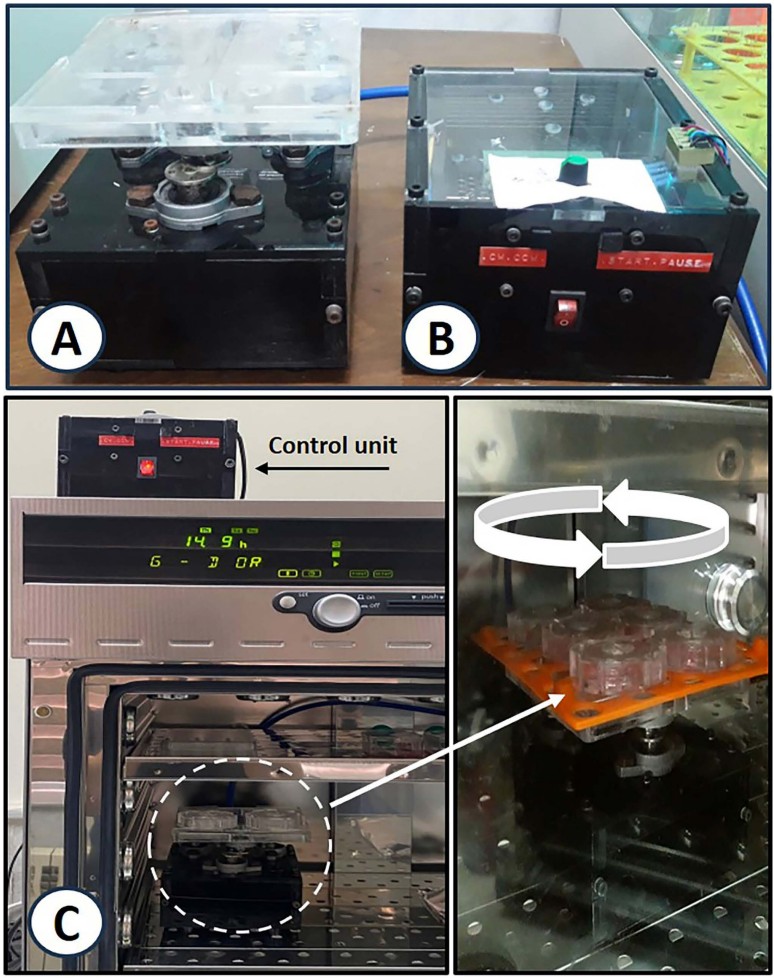

**Fig 2. Set-up of the perfusion-based co-culture device (POCC). (A)** The in-house developed rotator. **(B)** The control unit of the rotator. **(C)** Bioreactors were mounted on the rotator upper plate, and the entire unit (i.e., rotator and bioreactors) was placed inside an incubator. (The cultures were maintained at 37 °C, 5% CO2, and 95% humidity.

processes were performed using a vacuum oven (AFE200LV-60DH, ATRA, Iran) coupled with a vacuum pump (VE135 N, VALUE, China).

## 2.4. Chip characterization and optimization

### 2.4.1. Characterization of perfusion speed.
Perfusion of the culture medium through the cell culture chambers is achieved by rotating the bioreactor chip using the rotator. To measure the flow rate produced upon chip rotation to determine the proper speed and calculate the flow-induced shear stress, a method based on tracking the dye colour was developed. The interconnecting channels and bioreactors were filled with non-coloured distilled water (1200 μL). Then, 10 μL of the dye solution was gently added to one of the bioreactors. The rotator was then actuated to induce movement of the liquid inside the chip. The distribution of the dye was monitored and imaged at specific intervals. The time required for uniform dye distribution within the channel was recorded. The flow speed is the division of the total volume of the liquid inside the chip and the time. Flow speed measurements were repeated for three different concentrations of the dye solution to ensure that the dye concentration was not effective in determining the flow speed.

**2.4.2. Numerical simulation.** To achieve a better understanding of the flow distribution during streamlines and the applied shear stress (ASS) due to the flow of culture medium on cells, a numerical simulation was carried out in COMSOL Multiphysics 5.2. A laminar steady flow field, along with mass flow rate inlet and outlet conditions, were considered for the simulations. The tops of the channels and bioreactors were open; therefore, the slippery wall condition was selected as the corresponding boundary condition. The other boundaries were set as the no-slip wall boundary conditions.

## 2.5. Cell culture

Both HepG2 and 3T3-L1 cells were cultivated in T25 flasks containing high-glucose DMEM supplemented with 10% (v/v), FBS and 1% (v/v) penicillin–streptomycin (10,000 U/mL) and kept in a standard tissue culture incubator at 37 °C and 5% $CO_2$. Confluent cells (approximately 90%) were cultured by standard trypsin dissociation, and the culture medium was renewed every 2–3 days. No differences in the basal culture conditions were observed between the two cell types. 3T3-L1 cells were used in the pre-adipocyte state and were not differentiated into mature adipocytes before drug exposure.

## 2.6. Cytotoxicity assay

In the POCC, the best experimental conditions for viable cells were established through pilot and optimization studies, which determined the appropriate concentrations of Olz, CGA, and Met. For this purpose, stock solutions of Olz, CGA, and Met (100 mM) were prepared in dimethyl sulfoxide (DMSO). Working concentrations for cell treatment were prepared by performing serial dilutions in a complete culture medium.. Dilutions were made such that the final DMSO concentration in the diluted solutions was less than 1%. Briefly, 3T3-L1 and HepG2 cells were inserted in 96-well plates at an initial density of 3000 cells per well. Following a 24-hour incubation period, the cells were exposed to Olz, CGA, and Met for four days at concentrations ranging from 31.75 to 1000 μM [21,42]. A complete medium containing 0.1% DMSO was used as the untreated control. Viability was assessed using the AlamarBlue® method. The final concentrations of CGA and Olz, based on the preliminary studies performed for determination, were defined as 350 μM and 100 μM, respectively. To study the synergistic action of Olz and CGA, the cells were cultured in 96-well microplates and challenged with 350 μM CGA for ~1 h. The cells were then treated with 100 μM Olz and incubated for 4 days. However, this combination significantly lowered the cell survival rate. Therefore, further experiments were conducted using lower concentrations of 2.5–50 μM Olz, CGA, and Met, as they improved maintenance of cell viability with preserved efficacy. All assays were repeated in triplicate, and the final concentration of each compound was 50 μM. Prior to cell seeding, multiple microfluidic devices were fabricated and pre-tested using deionized water for at least 24 h to assess their structural integrity and potential leakage. Only devices that showed no signs of leakage were selected for cell seeding and subsequent experiments. Throughout the cell culture period, the devices were routinely monitored for leakage or delamination. If any structural failure was observed, the affected device was excluded from the study and replaced with a pre-tested backup device prepared for the corresponding experimental group. Only data obtained from intact and leak-free devices were included in the final analysis.

## 2.7. Experimental procedure

Whole chips were sterilized with 70% EtOH, cleaned three times with PBS under sterile conditions, and then exposed to UV light for 45 min before cell seeding in the bioreactors. To promote cell adhesion on the PMMA surface, 50 μL of FBS was added to each chamber of the chip and incubated overnight at 37 °C and 5% $CO_2$. Subsequently, 50 μL of 3T3-L1 and HepG2 suspensions were cultivated in the four chambers of the bioreactor chip at a density of $3 \times 10^3$ cells/well. The chips were then kept under static conditions at 37°C in a 5% $CO_2$ incubator for 48 h to form a confluent adherent cell layer before the flow was supplied. Finally, the potential of the POCC model in drug screening was investigated. The current study utilized three drugs: olanzapine, an antipsychotic drug that induces metabolic dysfunction; metformin, an

antidiabetic agent; and chlorogenic acid, a natural polyphenol for protection against metabolic abnormalities. Drug incubation was initiated 48 h after the beginning of the cell culture within the bioreactors.

Six different treatment conditions were performed including (i) Control (Cells + culture medium containing 0.1% DMSO), (ii) Olz alone (50 μM), (iii) CGA alone (50 μM), (iv) Met alone (50 μM), (v), and (vi) co-treatment with CGA (50 μM) or Met (50 μM), and Olz (50 μM). The test for drug exposure was extended to 4 days based on our pilot study that cells could be maintained on-chip without medium replacement. This timeframe for evaluating the treatment effects and metabolic alterations in various cell types is supported by prior studies [21,42]. No complete medium exchange was performed during the culture period of the study. In a limited number of cases, approximately 10% (120 μL) of the total medium volume was carefully removed and replaced with fresh medium after 48 h. This partial replenishment strategy was employed to support cell viability while minimizing the dilution of accumulated metabolic signals.A control experiment was also conducted in parallel in a 96-well plate as a monoculture and static model, using a cell density and drug concentration similar to those of the perfusion-based co-culture chip. The schematic representation of the experimental procedure is illustrated in Fig 3.

## 2.8. Cell viability

Cell viability was assessed by the AlamarBlue® method after four days of incubation. The culture medium was collected and replaced with fresh medium, and 10% AlamarBlue® was added to each well. Four hours later, cell viability was read at 570 and 600 nm using a microplate reader (Tecan, Infinite® M200 PRO, USA). We determined the decrease in cytotoxicity and proliferation between the treated and control cells by the cell viability [43]:

$$Percentage\ of\ viabilty = \frac{(O2 \times A1) - (O1 \times A2)}{(O2 \times P1) - (O1 \times P2)} \times 100$$

Where:
  O1 = molar extinction coefficient of oxidized AlamarBlue® (blue) at 570 nm that is 80586
  O2 = molar extinction coefficient of oxidized AlamarBlue® at 600 nm that is 17216
  A1 = absorbance of test wells at 570 nm
  A2 = absorbance of test wells at 600 nm
  P1 = absorbance of positive growth control well (cells plus AlamarBlue® but no test agent) at 570 nm
  P2 = absorbance of positive growth control well (cells plus AlamarBlue® but no test agent) at 600 nm

## 2.9. Oil red O (ORO) staining

To measure the impact of the medication on lipid accumulation within cells, ORO staining was performed in accordance with the manufacturer's instructions. A stock solution of ORO was prepared with 0.5 g/100 ml in 100% (v/v) isopropanol. For fresh preparation of the ORO working solution, the stock solution was diluted in a 6:4 or 3:2 ratio with ddH2O, incubated at room temperature for 10 min, and filtered using Whatman No. 1 [44]. Following four days of treatment, the medium was removed, the cells were twice washed with ice-cold PBS, and they were fixed in 10% neutral formalin for 30 min at room temperature to visualize intracellular lipids. Then the formalin was removed, and the cells were gently washed twice with dH2O. Next, each well was filled with 60% v/v isopropanol and maintained at room temperature for 5 min After that, isopropanol was removed and replaced with the ORO working solution. The solution was then incubated for 30 min at room temperature. Subsequently, ORO was removed, and the wells were washed three times with dH2O. Finally, the cells were coated with water, and images were taken under a microscope (Olympus AX70, Olympus Italia, Milan, Italy). For quantitative analysis, ORO was eluted with isopropanol 60% (v/v), and intracellular dye was extracted by adding isopropanol 100% (v/v) and incubating for 5 min at room temperature. Then, the OD was measured at 492 nm by a microplate reader (Tecan, Infinite® M200 PRO, USA) [45,46]. Lipid content was normalized to that of the control group and expressed as a percentage of the total content.

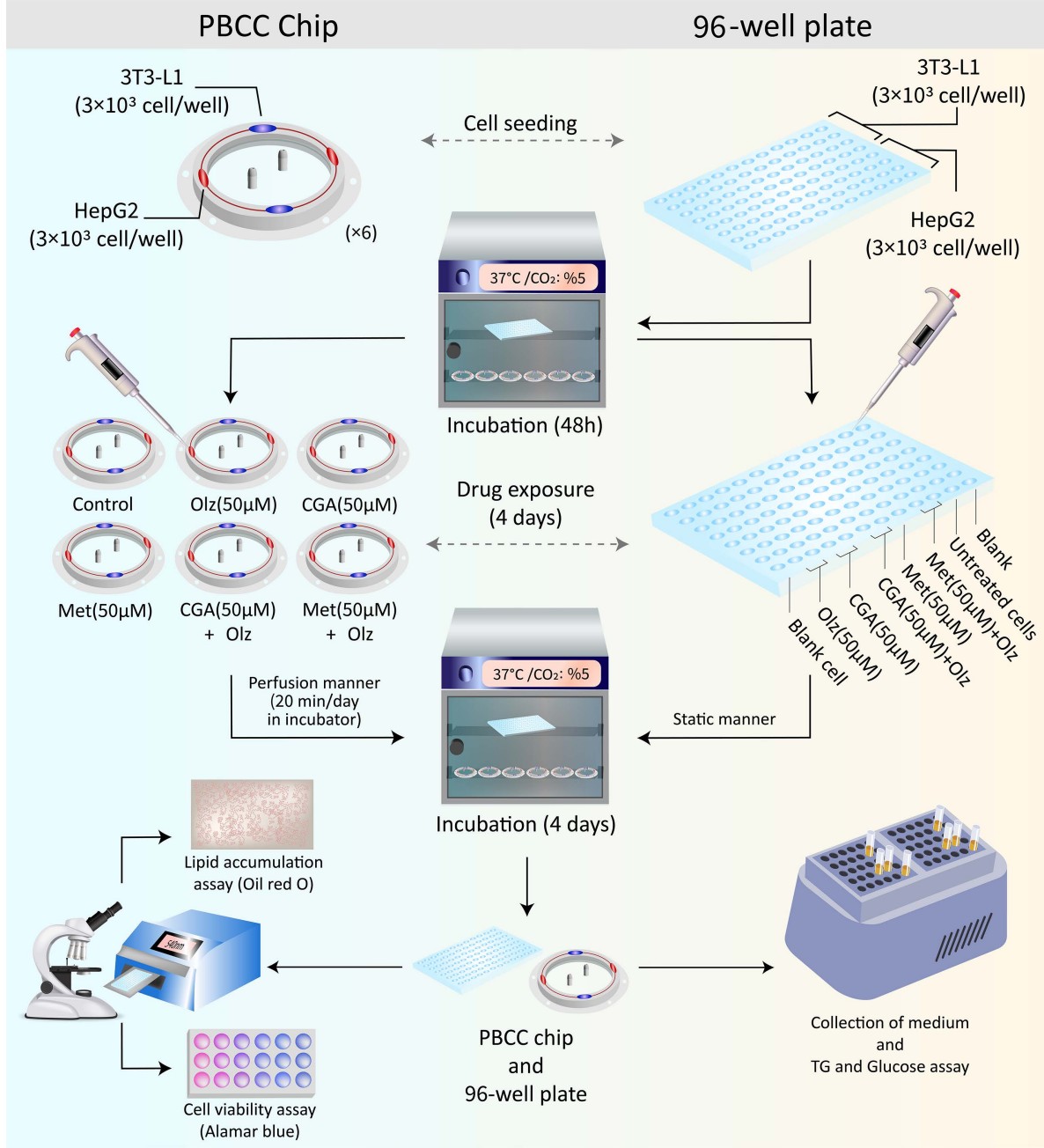

**Fig 3. The Schematic illustration of the experimental procedure.**

## 2.10. Statistical analysis

Experimental data represent the mean ± SEM, in triplicates. GraphPad Prism 8.0 software (GraphPad Prism Software Inc., San Diego, CA, USA) was used for all the statistical analyses. One-way ANOVA was performed, followed by the Tukey-Kramer multiple comparison post-test. P-values are considered statistically significant when below 0.05.

## 3. Results

### 3.1. Platform design and fabrication

In our POCC device, the rotation-induced fluid movement facilitated the continuous exchange of cell-secreted factors between the chambers. This rotation ensures adequate solute distribution through cell chambers (Fig 1-2). To assess the impact of the rotation speed, the co-culture of 3T3-L1 and HepG2 cells at 30 and 60 rpm was examined. The case of 30 rpm showed no detachment or damage to the cell morphology, while the higher speed led to adverse effects on cell attachment on the surfaces. The 30-rpm rate was selected for further experiments. The numerical simulation and cell culture procedure showed that the channel dimensions and chamber geometry carefully facilitated the continuous flow of the culture medium. The geometry of the open-top channels and bioreactors enabled sufficient perfusion of the culture medium upon rotation of the bioreactor chip, eliminating issues such as blockage, air bubble trapping, and disruption. Each chamber had dimensions equal to those of a single cavity in a standard 96-well microplate. Our standardized chamber format allows for an easy comparison between traditional static cell cultures and a POCC chip that can be subjected to fluid flow for studying, in particular, the effects of perfusion in a co-culture device. A cap was designed to cover the chip, allowing for gas exchange and preventing the evaporation of the culture medium and contamination. The four-chamber chip supports the growth of additional cell types, which facilitates convenient handling of cell seeding, imaging, and sampling procedures.

### 3.2. Flow characterization and modelling

The corresponding flow rate inside the chip was calculated to be 1.253 µL/s at a rotation speed of 30 rpm. The total liquid volume of the bioreactor was 1200 µL. At a rotation speed of 30 rpm, complete homogenization or uniform distribution of the solution in the chip was achieved after 20 min of operation, as shown in Fig 4A. A numerical simulation of the velocity profiles, flow lines, and shear stress through the bioreactors was performed based on the experimentally obtained flow velocities (Fig 4B). The COMSOL simulation in Fig 4E shows that the fluid circulated effectively within the bioreactor, and the mass transfer occurred at a designated rotation speed of 30 rpm. Fig 4C-4D illustrate the distribution of shear stress over the bottom of the bioreactors on which the cells were adhered. A flow-induced shear stress with an average value of $5.7 \times 10^{-6}$ Pa was obtained. The design of the deep culture chamber reveals that low shear stress on the adhered cells can reduce the possibility of cell detachment and enhance cell viability.

### 3.3. Cytotoxicity assays

The viability of 3T3-L1 and HepG2 cells from different groups was compared in 96-well plates and POCC chips. The dosages of chemicals required for the simultaneous treatment of cell lines with different sensitivities or resistances were optimized according to the literature [47,48]. The criterion was to minimize cell stress in sensitive lines while ensuring efficacy in resistant lines, thereby maximizing their viability and reproducibility [49]. We determined the IC50 of Olz and CGA in both cell types and performed a pilot test based on IC25, as shown in Table 1. Treatment of 3T3-L1 cells with Olz and CGA at the IC25 concentration induced cytotoxic effects within a well plate, but completely led to cell death in the POCC chip. Thus, we explored the concentration range of 2.5–50 µM for both Olz and CGA. Further experiments were performed based on the results obtained at a concentration of 50 µM. No significant difference in cell viability was found after four days in the 96-well plate and co-culture chip groups, as shown in Fig 5. The co-culture chip group only showed a trend toward lower cell viability than the well plate group (Fig 5A-5D). In addition, there were no changes in the appearance of the cells before and after incubation during this period.

### 3.4. Mimicking metabolic dysfunction with Olz on POCC

To evaluate the potential of the POCC chip as a tool for screening drug-induced metabolic disorders, Olz, CGA, and Met were selected as the model compounds. ORO staining provides both a visual and quantitative overview of cellular lipid

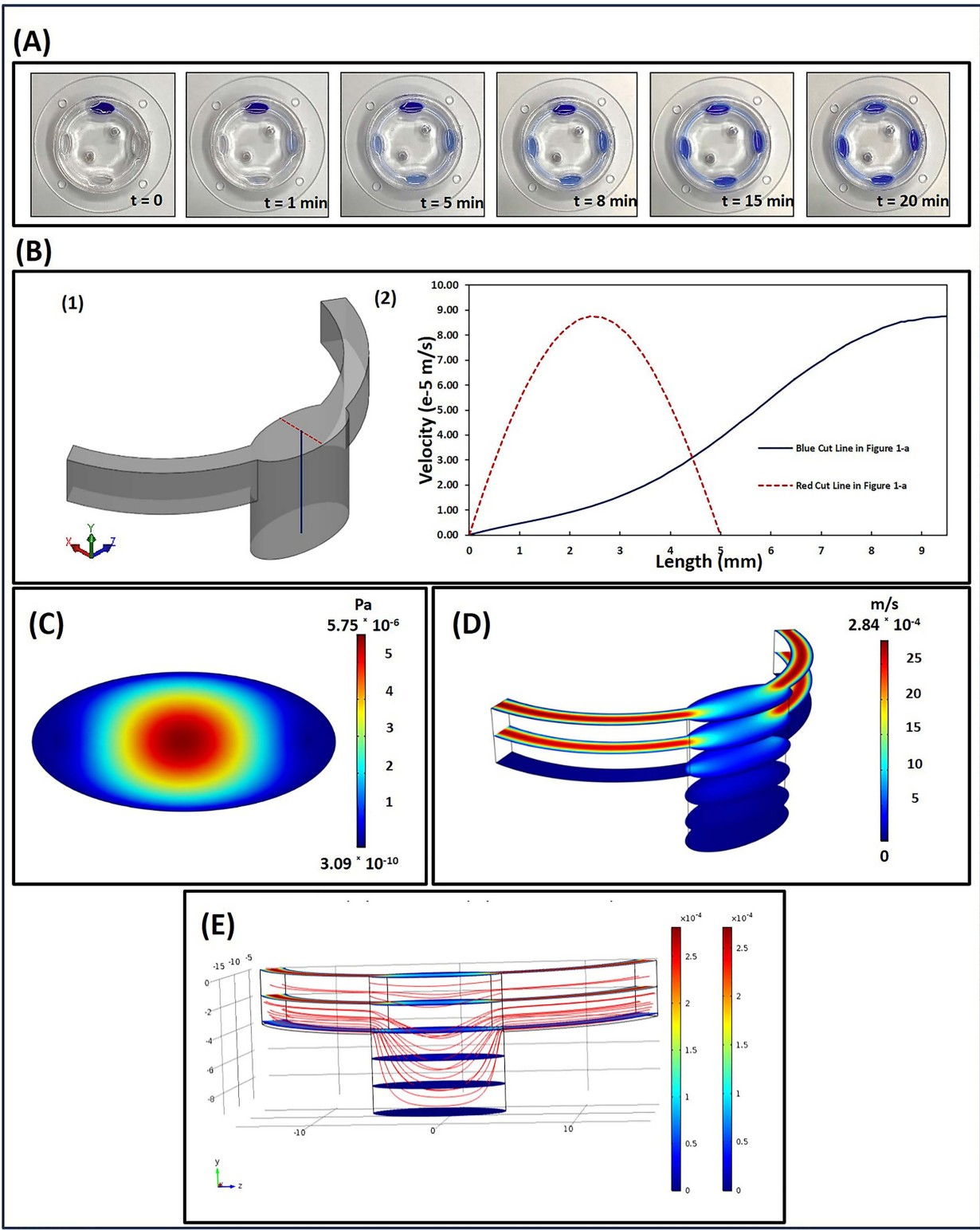

**Fig 4. Chip operation and numerical simulation. (A)** Spatiotemporal movement of dye inside the chip located on the rotator for rotation at the speed of 30 rpm. **(B-1)** The geometry of the bioreactor used for the applied shear stress (ASS) simulation. **(B-2)** Fluid velocity profiles inside the bioreactor in two cut lines have shown in Figure B-1. **(C)** The distribution of ASS over the bottom of the bioreactor where cells are adhered on. **(D)** Fluid velocity contours at different distances from the bottom of the bioreactor for the rotation speed of 30 rpm. **(E)** Streamlines inside the chip, showing how fluid moves..

**Table 1. IC50 and IC25 values (µM) for olanzapine (Olz), chlorogenic acid (CGA) and metformin (Met) in 3T3-L1 and HepG2 cell lines.**

| Cell lines | 3T3-L1 | | | HepG2 | | |
|---|---|---|---|---|---|---|
| Reagents | Olz | CGA | Met | Olz | CGA | Met |
| IC50 | 165.2 | 569.7 | >1000 | 230.71 | 903.47 | >1000 |
| IC25 | 83.32 | 306.13 | | 129.59 | 474.29 | |

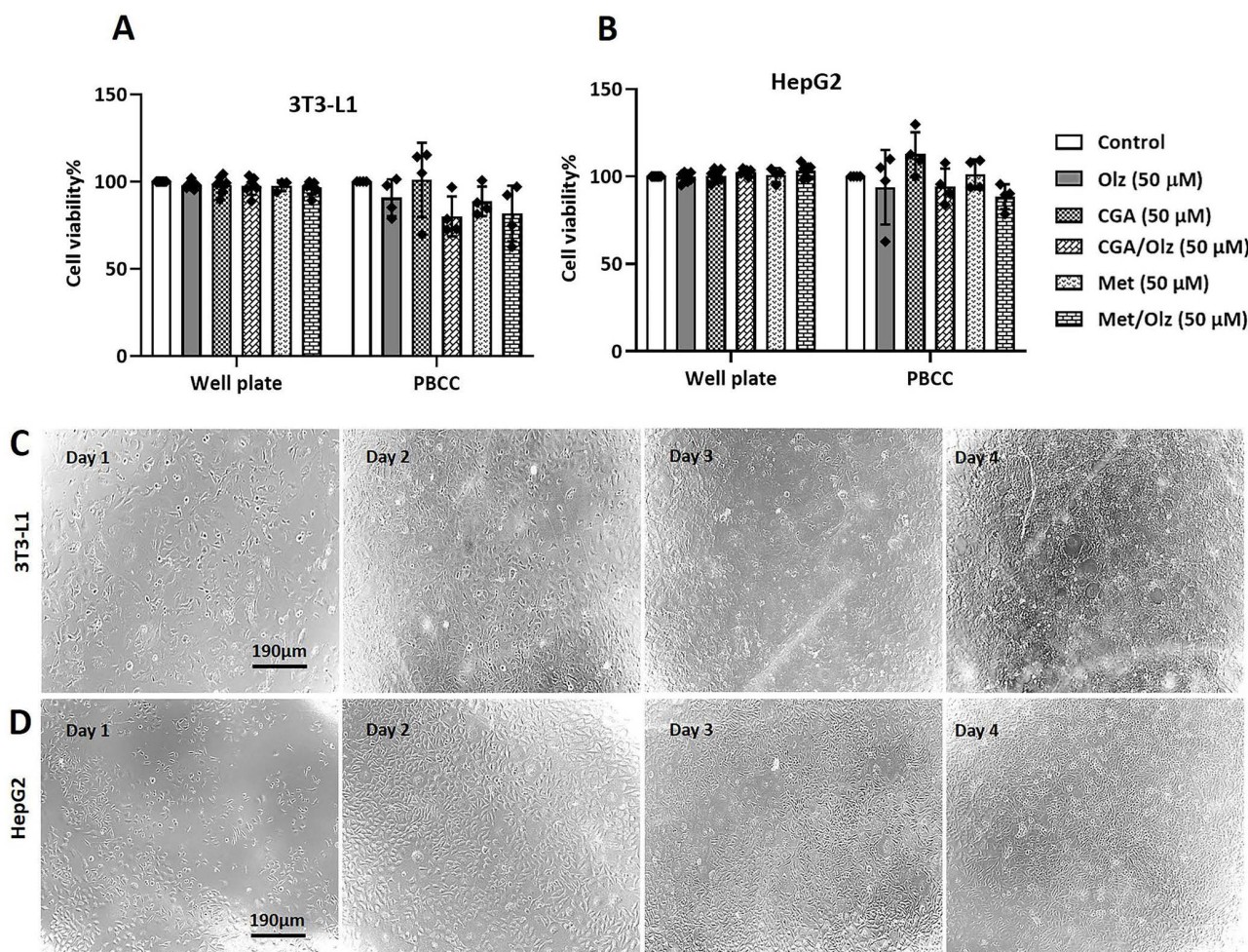

**Fig 5. Evaluating the cytotoxic effect of different treatment conditions following 4 days of exposure on (A) 3T3-L1 and (B) HepG2 cells in 96-well plates and the perfusion-based co-culture (POCC) device.** Cells were treated with olanzapine (Olz; 50 µM), chlorogenic acid (CGA; 50 µM), and metformin (Met; 50 µM) alone or in combination for 4 days. The AlamarBlue reduction was quantified and expressed as a viability percentage relative to the control (set at 100%). One-way ANOVA and Tukey's post-hoc tests were used for statistical analysis, and the data are shown as the mean±SEM for four independent biological experiments. No statistically significant differences in cell viability were observed between the treatment groups in either the 96-well plate or the co-culture system, nor when comparing the corresponding groups between the two systems. Microscopic images of growth and morphology changes of **(C)** 3T3-L1 and **(D)** HepG2 cells in POCC chips were captured during the experiment period using a microscope (Olympus, AX70, Olympus Italia, Milan, Italy). The images were taken at 20x magnification, with scale bars indicating 190 µm.

accumulation or reduction. Fig 6 shows microscopic images indicating changes in intracellular lipid content across the experimental groups in both cell lines, and the quantification data are shown in Fig 7. Olz significantly enhanced intracellular lipid accumulation in 3T3-L1 and HepG2 cells, in the POCC chip model, by ~30%±8% and ~63%±14%, respectively (Fig 7A-B).

Olz increased glucose concentrations by ~41%±6% (Fig 8A1). Although there was a trend toward elevated TG levels in the co-culture medium, this increase was not statistically significant (Fig 8B1). As for static culture in 96-well plates, Olz also significantly increased lipid accumulation, achieving levels of ~67%±16% in 3T3-L1 cells and ~19%±5% in HepG2 cells, Fig 7C-7D. Additionally, TG levels increased by ~31%±12% in HepG2 cells, while in 3T3-L1 cells, TG levels decreased by ~42%±9% (Fig 8B2-8B3). We also found that Olz reduced glucose by ~24%±7% in HepG2 cells (Fig 8A3).

These findings demonstrate that the POCC chip replicates the metabolic disruptions caused by Olz, including lipid accumulation and modified glucose/TG metabolism. This platform provides a more physiologically relevant readout than conventional static culture systems.

### 3.5. Anti-metabolic dysfunction drug testing on POCC

When CGA or Met was co-treated with Olz in our POCC model, lipid accumulation was significantly lower than that in the Olz group (Fig 7A-B). CGA and Met treatments exerted significant lipid-lowering effects in 3T3-L1 and HepG2 cells compared to the Olz group. In 3T3-L1 cells, CGA achieved a~63%±8% reduction and Met achieved a~35%±8% reduction, whereas in HepG2 cells, CGA decreased lipid content by ~69%±14% and Met by 59%±14% compared with the Olz control group. The CGA and Met treatments in HepG2 cells showed no statistical difference compared to those in the control group. CGA displayed profound inhibition of lipid accumulation, with ~35%±8% in 3T3-L1 cells. CGA also had a significant impact on glucose levels (Fig 8A1). According to POCC chip data, it was observed that CGA and Met alone caused no significant change in glucose levels, whereas CGA pre-treatment with Olz caused a decrease in glucose levels by ~36%±6%. Unexpectedly, Met, which acts as a glucose-reducing agent, did not lower Olz-induced high glucose levels in this study. In addition, while TG levels tended to decrease in the CGA and Met groups (Fig 8B1), these changes were not statistically significant compared with those in the Olz and control groups.

In static 96-well cultures, pretreatment of CGA with Olz also decreased lipid accumulation by ~68%±16% in 3T3-L1 cells, although Met did not significantly reduce Olz-induced lipid accumulation in this model (Fig 7C). CGA and Met alone had no significant effect on lipid accumulation compared with the control group in either cell line in the well plate (Fig 7C-7D). In monoculture settings, individual application of CGA in 3T3-L1 cells (~30%±7%) and Met in HepG2 cells (~23%±7%) significantly reduced glucose levels compared to their respective control groups (Fig 8A2-8A3). However, when used as a pretreatment before Olz exposure, neither CGA nor Met elicited any significant changes in glucose levels relative to the olanzapine-treated group. In 3T3-L1 cells, neither CGA nor its combination with Olz significantly affected the TG levels (Fig 8B2). In contrast, in HepG2 cells, CGA pretreatment led to a significant reduction in TG levels by ~40%±12% compared to the Olz group and by ~51%±12% (Fig 8B3). Met significantly reduced TG levels in 3T3-L1 cells by ~31%±9%, but showed no significant effect when pre-administered with Olz. However, in HepG2 cells, Met, whether pre-administered with Olz or applied alone, did not significantly alter TG levels compared to the Olz-treated group and the control group, respectively. These observations suggest that CGA may exert a more substantial protective effect than metformin against the lipogenic effects of Olz.

## 4. Discussions

The current study found that our platform closely replicated the metabolic alterations caused by Olz, including increased lipid, glucose, and TG accumulation. Another important finding was that pre-treatment with CGA and Met within the chip significantly decreased lipid accumulation, glucose, and TG levels. Our pervious animal study showed that Olz increased serum glucose and TG levels and visceral fat, and CCA and Met reduced these metabolic effects in that study [50].

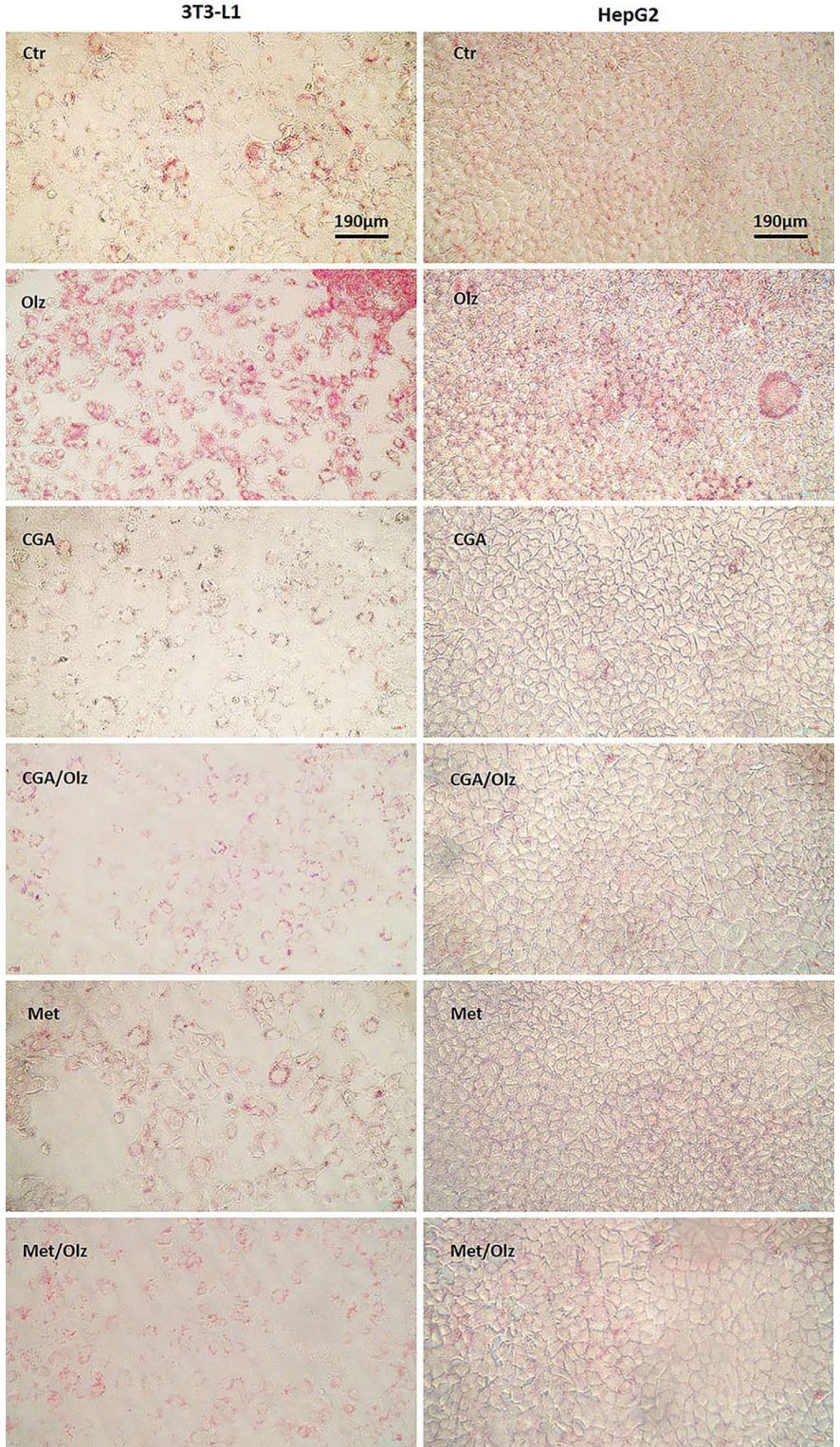

**Fig 6. Micro-images of intracellular lipid content that were stained with Oil Red O.** 3T3-L1 and HepG2 cells were treated with Olz (50 μM), CGA (50 μM), and Met (50 μM) alone or in combination for 4 days. Oil Red O-stained images were captured using the Nikon Eclipse Ti inverted microscope and NIS Elements imaging software at 20×magnification.

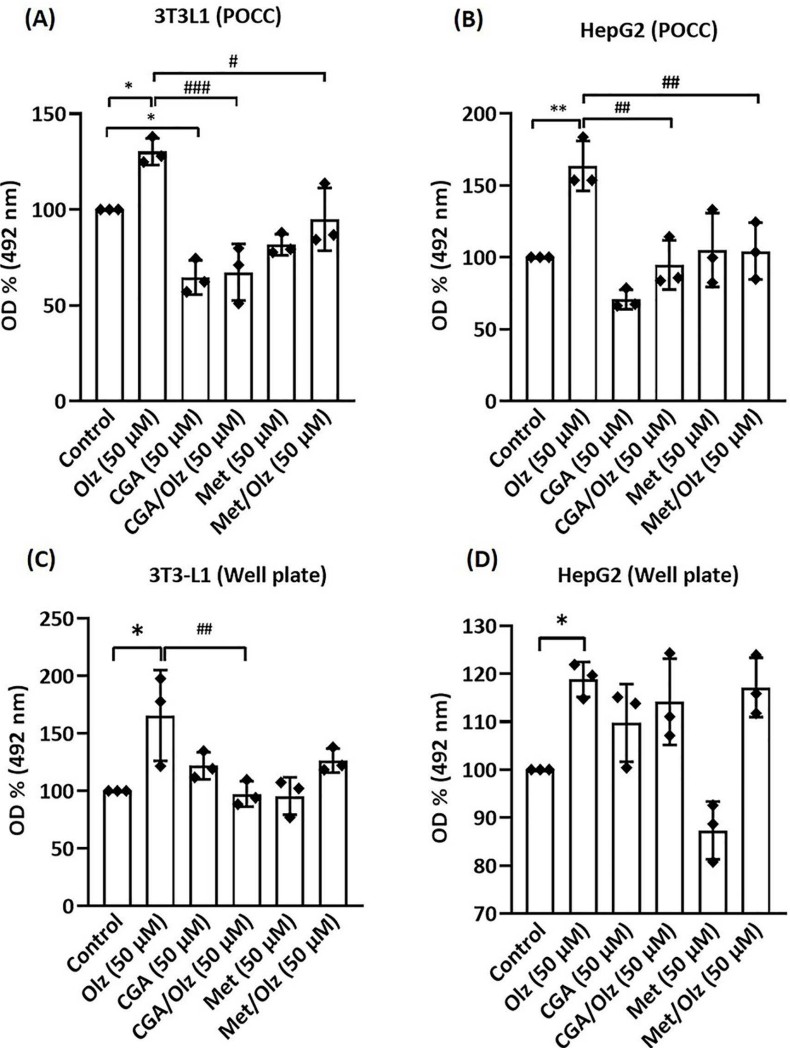

**Fig 7. Quantitative Oil Red O assay of the intracellular lipid content of different treatment conditions on 3T3-L1 and HepG2 cells in the perfusion-based co-culture (POCC) device (A and B) and in a 96-well plate (C and D).** Cells were treated with olanzapine (Olz; 50 μM), chlorogenic acid (CGA; 50 μM), and metformin (Met; 50 μM) alone or in combination for 4 days. Results are expressed as the percent ratio of optical density (OD) of the treated sample in comparison with control readings at 492 nm. The statistical differences between groups were compared by one-way ANOVA, and the data are shown as the mean ± SEM for three independent biological experiments, each conducted in triplicate. Statistical significance between groups is denoted as *$p < 0.05$ and **$p < 0.01$ vs. the control group, ## $p < 0.01$, #### $p < 0.0001$ vs. the olanzapine group.

According to these outcomes highlight that, compared with 2D static cultures, the POCC model more closely matches the prediction of the effects of drugs on lipid accumulation and glucose metabolism.

Cell–cell interactions, particularly between metabolically distinct tissues, have gained increasing attention because of their physiological relevance in both fundamental and translational biomedical research [51]. The POCC platform developed in this study consists of four interconnected culture chambers, enabling the simultaneous co-culture of different cell types and facilitating biochemical communication between tissue compartments.

Passive medium circulation was achieved using a rotator-based system, which allowed the controlled and homogeneous exchange of soluble factors between chambers while maintaining low shear stress conditions compatible with

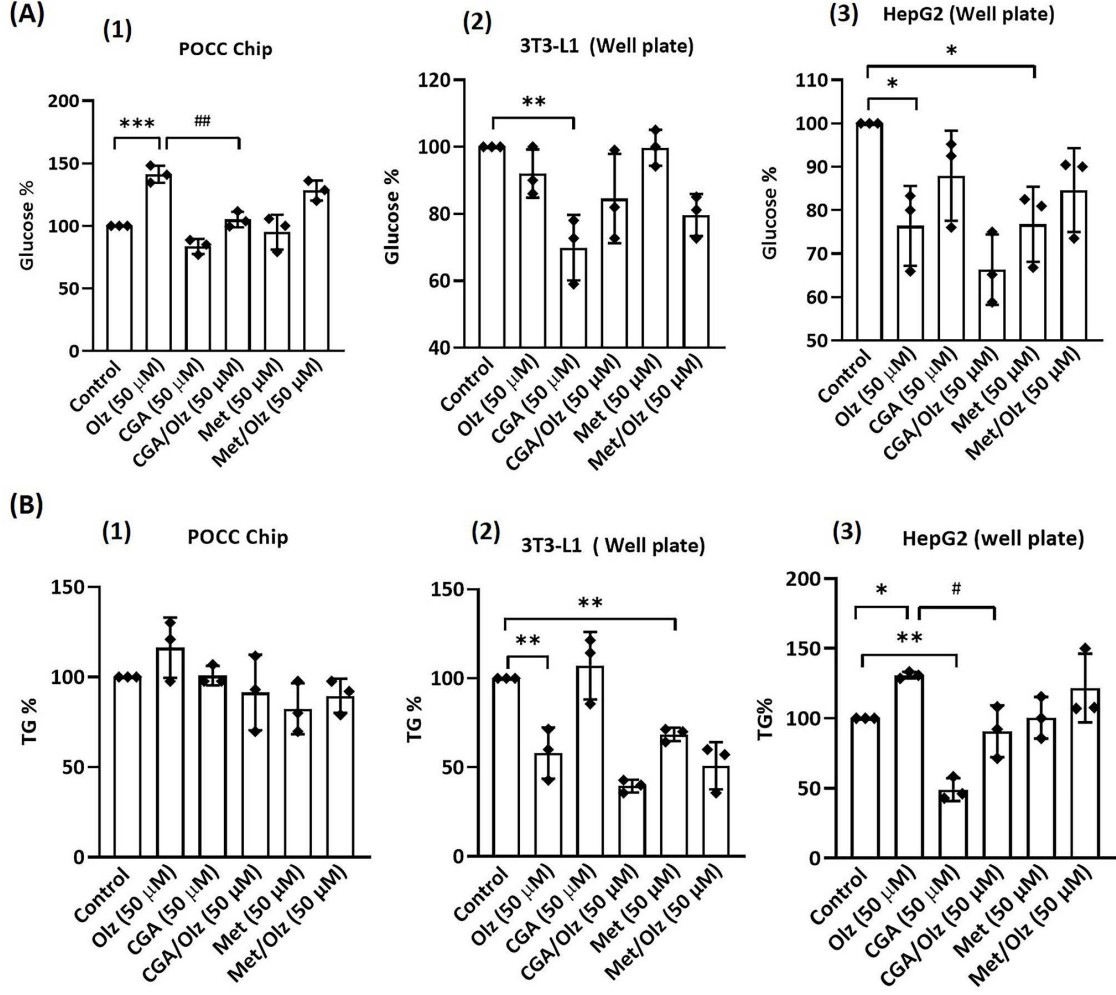

**Fig 8. Effect of different treatment conditions on medium metabolite concentrations over 4 days (no media exchange). (A)** Glucose concentration in the culture medium for **(1)** the perfusion-based co-culture (POCC) device, **(2)** 3T3-L1 cells, and **(3)** HepG2 cells in a 96-well plate. **(B)** Triglyceride (TG) concentration in the culture medium for **(1)** the POCC device, **(2)** 3T3-L1 cells, and **(3)** HepG2 cells in a 96-well plate. Cells were treated with olanzapine (Olz; 50 µM), chlorogenic acid (CGA; 50 µM), and metformin (Met; 50 µM) alone or in combination for 4 days. Concentrations were expressed as a percentage relative to the control (set at 100%), and data are presented as mean ± SEM from triplicate experiments. One-way ANOVA and Tukey's post-hoc tests were used for statistical analysis. * $p < 0.05$, ** $p < 0.01$, *** $p < 0.001$ vs. control group; # $p < 0.05$, ## $p < 0.01$ vs. olanzapine group.

cell viability. The geometric design of the channels and chambers ensured a uniform flow without obstruction, minimized bubble formation, and supported efficient gas exchange through the open-top configuration. Importantly, the chamber dimensions were designed to be comparable to those of a standard 96-well plate, enabling a direct comparison between monoculture and co-culture conditions. While the present study focused on liver–adipose interactions, the modular design of the platform allows scalability to additional tissue compartments, providing flexibility for modeling more complex inter-organ communication in future studies. Unlike conventional static co-culture systems, such as Transwell inserts, which rely on diffusion-limited exchange and lack dynamic medium circulation, the POCC platform enables continuous biochemical communication under controlled flow conditions, thereby improving the physiological relevance of inter-tissue metabolic interactions.

Metabolic disorders indicate abnormalities in glucose and lipid metabolism in various organs, especially the adipose tissue and liver [52]. The liver, as the primary site of unexpected adverse effects, regulates drug metabolism and controls drug half-life [38,53]. Interactions between liver and adipose tissues regulate metabolic balance by producing growth factors and adipokines [1,46].

Previously, we investigated the effect of CGA on Olz-induced metabolic syndrome in rats [50]. The data obtained from this animal study were used as references for the design and validation of the system. In this study, HepG2 and 3T3-L1 cells were used to investigate these connections Although physiological relevance is an important consideration in multi-tissue *in vitro* models, translating *in vivo* hepatic–adipose tissue proportions into a fixed *in vitro* cell ratio remains challenging [54]. Adipose tissue mass and metabolic activity are highly dynamic and vary substantially according to nutritional status, body composition, and disease state. Different adipose depots contribute differently to the systemic metabolism. These factors complicate the definition of an *in vivo* tissue ratio and highlight that simple mass-based ratios may not fully reflect physiological dynamics [55,56]. Consequently, liver–adipose co-culture systems generally prioritize capturing paracrine metabolic communication rather than replicating the exact *in vivo* tissue proportions. In this study, a 1:1 ratio of 3T3-L1 to HepG2 cells was selected as a balanced and reproducible starting point to support bidirectional metabolic signaling between compartments while maintaining experimental simplicity and robustness in this proof-of-concept platform. In Future studies, other ratios of hepatic to adipose cells could be investigated according to the scope of the study.

Several studies have revealed that Olz is linked to severe metabolic adverse effects through alterations in lipid and glucose metabolism by directly affecting the liver and adipose tissue, as well as the central systems of energy balance [57–61]. According to *in vivo* studies, Olz increases serum TG and glucose levels, weight gain, and visceral fat [50,62,63], and has also been shown to induce adipogenesis in 3T3-L1 [64,65] and HepG2 cells *in vitro* [66,67].

In this study, Olz significantly increased lipid accumulation and glucose concentrations in both HepG2 and 3T3-L1 cells in the POCC model, which closely mimicked the metabolic dysfunction observed in clinical and preclinical studies [9,50].

*In vitro* findings suggest that CGA plays a significant role in reducing the proliferation of 3T3-L1 preadipocytes and inhibiting lipid accumulation during adipocytic differentiation of 3T3-L1 cells [11,68,69]. Additionally, clinical and animal studies have revealed that CGA may be advantageous in the management of obesity and metabolic disorders [10,70,71]. In animal studies, CGA has been found to reduce free fatty acids, TG, and plasma cholesterol. It also increased the HDL-cholesterol/total cholesterol ratio compared to that of the high-fat control group [72]. Our previous study demonstrated that CGA supplementation effectively mitigates Olz-induced weight gain and metabolic abnormalities by modulating the hypothalamic pathways associated with appetite regulation. Specifically, CGA reversed the Olz-induced upregulation of NPY and pAMPK and downregulation of POMC expression. The efficacy of CGA was found to be comparable to that of Met in addressing Olz-induced metabolic syndrome, suggesting its potential as a promising supplement [50].

Prior studies have also demonstrated that Met alleviates insulin resistance, diminishes lipid accumulation, and enhances glucose homeostasis in rodent models of Olz-induced metabolic syndromes [7,9,67,73–75]. However, our data showed that CGA likely offers better protection, particularly by reducing lipid accumulation.

In line with previous studies, the POCC results showed that co-treatment with CGA and Olz led to a greater reduction in lipid content than that with Met co-treatment, with a ~28% decrease in 3T3-L1 cells and a ~10% decrease in HepG2 cells.

Using just CGA resulted in 17% and 24% decrement in lipid accumulation in comparison with Met treatment in 3T3-L1 and HepG2 cells, respectively. These data identified that CGA was more effective than Met to decrease dramatically lipid accumulation. This result referred to as more evidence for CGA's potential as a therapeutic agent and goes along with our recent study on effects of CGA on Olz-induced metabolic dysfunction [50]. Given the observed changes in glucose, TG, and lipid accumulation levels in this study, would suggest that Olz, CGA, and Met may have a great effect on critical metabolic pathways. Previous studies have reported that Olz induces adipogenesis through the SREBP-1 pathway in 3T3-L1 cells, leading to increased lipogenesis and lipid deposition in these cells [65]. It also gets involve impairment in insulin signaling to have increment in glucose and lipid levels [60]. A recent study demonstrated that CGA and its isomers

effectively mitigated NAFLD in HepG2 cells. This effect was achieved by reducing lipid accumulation and TG levels [76]. CGA has been reported to exert anti-lipogenic effects in 3T3-L1 adipocytes through AMP-activated protein kinase (AMPK) activation. What is more prevention of fatty acid synthesis via suppression of acetyl-CoA carboxylase and peroxisome proliferator-activated receptor gamma (PPARγ) [76,77]. The mechanism of metformin also involves AMPK activation, which enhances glucose uptake and inhibits lipogenesis [8].

One of the notable aspects of this study was the direct comparison between cellular responses obtained from conventional static 96-well plate cultures and the POCC microfluidic platform. In static 96-well cultures, Olz-, CGA-, and Met-induced changes in lipid, glucose, and triglyceride accumulation were heterogeneous across individual cell types and, in some cases, even contradictory, showing limited concordance with the patterns reported in *in vivo* models [50]. In contrast, in the microfluidic chip system, the observed alterations in intracellular glucose and lipid levels were more consistent with the findings from animal studies. These findings suggest that in the current POCC system, intercellular communication facilitated by interconnected channels and intermittent passive perfusion under low shear stress may enhance physiological relevance by improving metabolite distribution and exchange between culture compartments. This configuration allows soluble factors secreted by hepatic and adipose cells to circulate between compartments without substantial washout or dilution.

Although this system does not replicate vascular perfusion or hepatic sinusoidal architecture, the low-shear, periodic circulation of the culture medium may partially support a more uniform exposure of hepatocytes to soluble metabolic signals, thereby approximating certain aspects of the hepatic metabolic microenvironment observed *in vivo*. Moreover, the compartmentalized configuration enables hepatic and adipose cells to maintain tissue-specific phenotypes while supporting paracrine and endocrine-like communication through shared medium circulation. Such functional coupling more closely approximates the liver–adipose metabolic crosstalk observed *in vivo*, even in the absence of continuous perfusion or endothelial-lined vasculature. Consequently, drug-induced metabolic responses in the POCC system, particularly changes in intracellular lipid accumulation and glucose handling, were more physiologically aligned with *in vivo* observations than those obtained under static culture conditions.

Albumin secretion and urea synthesis are widely accepted liver-specific functional markers for validating hepatic synthetic activity *in vitro* liver models and under dynamic and perfused culture conditions in liver-on-chip systems [78]. Such measurements are commonly incorporated into later-stage validation studies once platform feasibility, stability, and metabolic responsiveness have been established.

In the present study, the primary objective was not to establish a fully validated liver disease model but rather to develop and apply a pumpless co-culture platform to investigate inter-tissue metabolic communication and drug-induced metabolic alterations between the liver and adipose compartments. Accordingly, direct functional validation assays of hepatic synthetic activity, such as albumin and urea production, were not included and are acknowledged as limitations of the current study. Future studies will incorporate these established functional endpoints to further validate hepatic performance within the POCC system, as demonstrated in previous perfused HepG2-based microfluidic models [79].

Another limitation of the present study is that 3T3-L1 cells were used in the pre-adipocyte stage rather than fully differentiated adipocytes. Although this approach enabled the investigation of early lipid accumulation and liver–adipose crosstalk within the microfluidic platform, it limited conclusions regarding mature adipocyte-specific functions, such as adipokine secretion.

Notably, previous studies have shown that hepatocyte-derived conditioned media and extracellular vesicles can influence adipogenic and lipogenic responses in 3T3-L1 preadipocytes,even in the absence of standard differentiation protocols [80]. Although EV-mediated signaling was not directly assessed in the present study, such paracrine mechanisms may contribute to the enhanced lipid accumulation observed in the co-culture system and warrant further investigations. In conclusion, this innovation platform is a promising device for a massive reduction in dependence on animal testing, which improves the predictive performance regarding potential toxicity.

## Supporting information

**S1 File. Supporting information is available as a separate file via hyperlinks in the Supporting Information section of the article.**
(DOCX)

## Acknowledgments

The authors appreciate consultancy of Dr. Amir K. Miri during conceptualization and drafting this work. Technical comments from the Clinical Research Unit of Ghaem Hospital are also acknowledged.

## Author contributions

**Conceptualization:** Zeinab Ebrahimian, Hossein Hosseinzadeh, Bibi Marjan Razavi.

**Data curation:** Zeinab Ebrahimian, Amir Reza Ameri.

**Formal analysis:** Zeinab Ebrahimian, Fatemeh Kalalinia, Amir Reza Ameri.

**Funding acquisition:** Hossein Hosseinzadeh.

**Investigation:** Zeinab Ebrahimian, Amir Reza Ameri.

**Methodology:** Zeinab Ebrahimian, Fatemeh Kalalinia, Amir Reza Ameri, Seyed Ali Mousavi Shaegh.

**Project administration:** Fatemeh Kalalinia, Hossein Hosseinzadeh, Seyed Ali Mousavi Shaegh.

**Resources:** Hossein Hosseinzadeh, Seyed Ali Mousavi Shaegh.

**Software:** Amir Reza Ameri.

**Supervision:** Fatemeh Kalalinia, Hossein Hosseinzadeh, Bibi Marjan Razavi, Seyed Ali Mousavi Shaegh.

**Validation:** Seyed Ali Mousavi Shaegh.

**Visualization:** Zeinab Ebrahimian, Amir Reza Ameri.

**Writing – original draft:** Zeinab Ebrahimian.

**Writing – review & editing:** Fatemeh Kalalinia, Amir Reza Ameri, Hossein Hosseinzadeh, Bibi Marjan Razavi, Seyed Ali Mousavi Shaegh.

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
