## [Decision Letter · Decision Letter 0]

6 Jan 2026

PONE-D-25-60517A pumpless liver-adipose model for studying metabolic dysfunction and drug responsesPLOS One

Dear Dr. Mousavi Shaegh,

Thank you for submitting your manuscript to PLOS ONE. After careful consideration, we feel that it has merit but does not fully meet PLOS ONE’s publication criteria as it currently stands. Therefore, we invite you to submit a revised version of the manuscript that addresses the points raised during the review process.

If applicable, we recommend that you deposit your laboratory protocols in protocols.io to enhance the reproducibility of your results. Protocols.io assigns your protocol its own identifier (DOI) so that it can be cited independently in the future. For instructions see: https://journals.plos.org/plosone/s/submission-guidelines#loc-laboratory-protocols. Additionally, PLOS ONE offers an option for publishing peer-reviewed Lab Protocol articles, which describe protocols hosted on protocols.io. Read more information on sharing protocols at . Additionally, PLOS ONE offers an option for publishing peer-reviewed Lab Protocol articles, which describe protocols hosted on protocols.io. Read more information on sharing protocols at https://plos.org/protocols?utm_medium=editorial-email&utm_source=authorletters&utm_campaign=protocols..

We look forward to receiving your revised manuscript.

Kind regards,

Santhi Silambanan, MD, DNB

Academic Editor

PLOS One

Journal Requirements:

“The authors are grateful to the Vice Chancellor of Research, Mashhad University of Medical Sciences (No. 971985), Mashhad, Iran for financial support”

3. We note that your Data Availability Statement is currently as follows: All relevant data are within the manuscript and in Supporting Information files.

Additional Editor Comments:

Authors need to respond to the all the queries raised by reviewers

Reviewers' comments:

Reviewer's Responses to Questions

**Comments to the Author**

1. Is the manuscript technically sound, and do the data support the conclusions?

Reviewer #1: Partly

Reviewer #2: Yes

2. Has the statistical analysis been performed appropriately and rigorously? 

Reviewer #1: Yes

Reviewer #2: Yes

3. Have the authors made all data underlying the findings in their manuscript fully available?

Reviewer #1: Yes

Reviewer #2: Yes

4. Is the manuscript presented in an intelligible fashion and written in standard English?

Reviewer #1: Yes

Reviewer #2: Yes

5. Review Comments to the Author

Reviewer #1: The authors present a pumpless, open-top, passive-circulation co-culture device (POCC) composed of four chambers to simulate liver–adipose interactions. HepG2 and 3T3-L1 were cultured in separate compartments and exposed to drug treatments, with metabolic responses evaluated through viability, lipid accumulation, glucose consumption, and extracellular triglyceride (TG) levels. The platform is innovative and shows strong potential as a passive-flow liver–adipose screening system. With additional functional validation and clearer justification of experimental design, the work could become a valuable contribution to metabolic disease modeling and early drug response studies. The manuscript would be strengthened by addressing the following points:

Major Comments

Are there any leakage or delamination issues after device assembly or during perfusion? If so, how were these assessed and mitigated?

The authors should specify how frequently media were exchanged or replenished and justify that. Passive-flow metabolic models are sensitive to nutrient depletion and waste accumulation.

Section 2.5 states that cells were cultured in T25 flasks with high-glucose DMEM + 10% FBS + 1% Pen/Strep. The authors should confirm whether both cell types (HepG2 and 3T3-L1) were cultured under identical conditions, or justify any differences. Moreover, 3T3-L1 are pre-adipocytes. The authors should clearly state whether differentiation into mature adipocytes was performed before drug exposure. If not, this limits the metabolic relevance of adipokine and lipid-mobilization readouts.

The authors should justify why hepatocytes and adipocytes were cultured in separate chambers rather than together and having 4 representative chambers. If the intent is a liver–adipose interaction model, this should be explicitly stated earlier and supported with a proper rationale.

The 1:1 ratio of 3T3-L1 : HepG2 cells should be justified.

Before positioning the system as a disease model or an alternative to animal testing, the hepatic compartment should be validated for core liver metabolic/synthetic function under perfusion, even at a phenotypic/directional level. At minimum, I recommend assessing:

• Albumin secretion (e.g., in device vs static control in 96-well plate)

• Urea production trends

The authors should expand discussion on why passive-flow/perfused culture better mimics in vivo compared to static systems, ideally correlating the device phenotype with in vivo hepatic vascular and metabolic behavior.

In addition to bright-field microscopy, the study would benefit from live/dead fluorescence imaging (e.g., Calcein-AM + PI) to visually support viability under perfusion, as well as clearer microscopic evidence of morphology changes in each chamber in Figure 5.

Minor Comments

The paragraph describing serial dilution with <1% DMSO in Section 2.6 is repeated and should be removed for clarity.

Section 2.7 states that 3×10³ cells/well were seeded, while Figure 3 reports 3×10³ cells/mL. The correct units should be verified and standardized.

The authors reference Equation (1) for calculating flow rate, but no equation was found in the manuscript. This should be corrected.

Future Work Suggestions

Transitioning the co-culture into 3D spheroids would significantly improve physiological relevance. This could be achieved by:

o Non-adhesive well surface treatment

o Increased cell density

o Incorporation of ECM-based hydrogels to support 3D tissue organization.

Reviewer #2: The work is good. The author should include some genomic studies and in vivo studies in mouse model. If in vovo study in animal model is not possible then it may be discussed with appripriate work previously done

6. PLOS authors have the option to publish the peer review history of their article (what does this mean?). If published, this will include your full peer review and any attached files.). If published, this will include your full peer review and any attached files.

.

Reviewer #1: **Yes:** Hossein AbolhassaniHossein Abolhassani

Reviewer #2: No

---

## [Author Response · Author response to Decision Letter 1]

24 Feb 2026

Dear Editor-in-Chief,

We thank the editor and the reviewers for their careful, constructive, and insightful comments, which significantly improved the quality and clarity of the manuscript. All comments and queries were responded in a point-by-point manner, and the corresponding revisions have been incorporated into the revised manuscript and highlighted using Track Changes.

Below, we provide a detailed, point-by-point response to all reviewer comments.

Editor:

Response: We have reviewed the manuscript in detail and confirmed that it adheres to the PLOS ONE formatting and style guidelines, as outlined in the provided templates. The manuscript files have been prepared accordingly.

2. Thank you for stating the following financial disclosure: “The authors are grateful to the Vice Chancellor of Research, Mashhad University of Medical Sciences (No. 971985), Mashhad, Iran for financial support”

Response: We confirm that the funders had no role in the study design, data collection and analysis, decision to publish, or preparation of the manuscript. We have included this statement in the cover letter as requested.

3. We note that your Data Availability Statement is currently as follows: All relevant data are within the manuscript and in Supporting Information files. Please confirm at this time whether or not your submission contains all raw data required to replicate the results of your study. Authors must share the “minimal data set” for their submission. PLOS defines the minimal data set to consist of the data required to replicate all study findings reported in the article, as well as related metadata and methods (https://journals.plos.org/plosone/s/data-availability#loc-minimal-data-set-definition)....

Response: Thank you for this clarification. The manuscript reports summary data (means and variability measures) based on raw data derived from experiments, in form of figures and table. No additional raw datasets beyond those reported were generated or analyzed. Raw data will be available upon request. If needed, please revise the Data Availability Statement to accurately reflect the data included in this study, as follows: “Data will be available by the authors upon a request.”

Response: The reviewers did not recommend citation of any specific previously published works; therefore, no additional references were required.

5. Additional Editor Comments: Authors need to respond to the all the queries raised by reviewers

Response: We carefully reviewed all reviewers’ comments. All reviewers’ queries were fully addressed, and corresponding revisions were made in the manuscript where appropriate.

Reviewer 1

1. Are there any leakage or delamination issues after device assembly or during perfusion? If so, how were these assessed and mitigated?

Response: We thank the reviewer for raising this important point. This procedure has been added to Section 2.6 of the revised manuscript (lines 195 and 201): “Prior to cell seeding, multiple microfluidic devices were fabricated and pre-tested using deionized water for at least 24 hours to assess structural integrity and potential leakage. Only devices showing no signs of leakage were selected for cell seeding and subsequent experiments. Throughout the cell culture period, devices were routinely monitored for leakage or delamination. If any structural failure was observed, the affected device was excluded from the study and replaced with a pre-tested backup device prepared for the corresponding experimental group. Only data obtained from intact, leak-free devices were included in the final analysis.”

2. The authors should specify how frequently media were exchanged or replenished and justify that. Passive-flow metabolic models are sensitive to nutrient depletion and waste accumulation.

Response: To preserve accumulated metabolic signals and prevent dilution of secreted factors, the culture medium was not routinely exchanged during the experiments. Preliminary optimization studies were performed to identify a cell seeding density that could maintain viability and metabolic activity over the four-day culture period without frequent medium replacement. Various seeding densities for HepG2 and 3T3-L1 cells were tested, and a density of 3,000 cells per compartment was selected as it ensured stable metabolic activity while avoiding nutrient depletion and excessive waste accumulation. To clarify this point, we have added the following description to Section 2.7 of the revised manuscript (lines 219 and 223): “No complete medium exchange was performed during the culture period. In a limited number of cases, approximately 10% (120 µl) of the total medium volume was carefully removed and replaced with fresh medium after 48 hours. This partial replenishment strategy was employed to support cell viability while minimizing dilution of accumulated metabolic signals.”

3. Section 2.5 states that cells were cultured in T25 flasks with high-glucose DMEM + 10% FBS + 1% Pen/Strep. The authors should confirm whether both cell types (HepG2 and 3T3-L1) were cultured under identical conditions, or justify any differences.

Response: Both HepG2 and 3T3-L1 cells were routinely cultured under identical conditions prior to seeding into the microfluidic devices, using high-glucose DMEM supplemented with 10% fetal bovine serum (FBS) and 1% penicillin–streptomycin. No differences in basal culture conditions were applied between the two cell types. This clarification has now been added to Section 2.5 of the revised manuscript (lines 168 and 172): “Both HepG2 and 3T3-L1 cells …. No differences in basal culture conditions were applied between the two cell types.”

4. 3T3-L1 are pre-adipocytes. The authors should clearly state whether differentiation into mature adipocytes was performed before drug exposure. If not, this limits the metabolic relevance of adipokine and lipid-mobilization readouts.

Response: We thank the reviewer for this important comment. In the present study, 3T3-L1 cells were used in the pre-adipocyte stage and were not differentiated into mature adipocytes prior to drug exposure. This was a deliberate design choice aimed at investigating early lipid accumulation and adipogenic responses of adipose precursor cells under the influence of hepatocyte-derived paracrine signaling within the co-culture system. Lipid accumulation was assessed using Oil Red O staining, a well-established marker of adipogenic and lipogenic activity in 3T3-L1 pre-adipocytes.

Increased triglyceride accumulation observed in 3T3-L1 pre-adipocytes exposed to hepatocyte-conditioned environments is consistent with previous reports demonstrating enhanced lipid accumulation in 3T3-L1 cells treated with hepatocyte-conditioned media under metabolic stress conditions (1). These findings support the relevance of the selected model for studying early liver–adipose metabolic crosstalk.

We acknowledge that the use of pre-adipocytes limits conclusions regarding mature adipocyte-specific functions. This limitation, together with relevant supporting literature, has now been explicitly discussed in the revised discussion section (lines 527-535): “Another limitation of the present study is that 3T3-L1 cells were used in the pre-adipocyte stage rather than fully differentiated adipocytes. While this approach enabled the investigation of early lipid accumulation and liver–adipose crosstalk within the microfluidic platform, it limits conclusions regarding mature adipocyte-specific functions such as adipokine secretion.

Notably, previous studies have shown that hepatocyte-derived conditioned media and extracellular vesicles can influence adipogenic and lipogenic responses in 3T3-L1 pre-adipocytes even in the absence of standard differentiation protocols [80]. Although EV-mediated signaling was not directly assessed in the present study, such paracrine mechanisms may contribute to the enhanced lipid accumulation observed in the co-culture system and warrant further investigation.” In addition, we have clarified in Section 2.5 of the Methods (lines 172-174): “3T3-L1 cells were used in the pre-adipocyte state and they were not differentiated into mature adipocytes prior to drug exposure.”

5. The authors should justify why hepatocytes and adipocytes were cultured in separate chambers rather than together and having 4 representative chambers. If the intent is a liver–adipose interaction model, this should be explicitly stated earlier and supported with a proper rationale.

Response: In the initial version of our model, hepatic and adipose cells were cultured in a simplified, compartmentalized configuration to establish a controllable and reproducible platform for studying inter-tissue metabolic communication. Accordingly, we have expanded the discussion section (lines 410-427) to further describe the design rationale and advantages of the POCC platform, particularly in terms of enabling dynamic inter-tissue communication, low-shear passive perfusion, and direct comparison with conventional static co-culture systems.

“Cell–cell interactions, particularly between metabolically distinct tissues, have gained increasing attention due to their physiological relevance in both fundamental and translational biomedical research [51]. The POCC platform developed in this study consists of four interconnected culture chambers, enabling simultaneous co-culture of different cell types and facilitating biochemical communication between tissue compartments.

Passive medium circulation was achieved using a rotator-based system, which allowed controlled and homogeneous exchange of soluble factors between chambers while maintaining low shear stress conditions compatible with cell viability. The geometric design of the channels and chambers ensured uniform flow without obstruction, minimized bubble formation, and supported efficient gas exchange through the open-top configuration. Importantly, the chamber dimensions were designed to be comparable to those of a standard 96-well plate, enabling direct comparison between monoculture and co-culture conditions. While the present study focused on liver–adipose interactions, the modular design of the platform allows scalability to additional tissue compartments, providing flexibility for modeling more complex inter-organ communication in future studies. Unlike conventional static co-culture systems such as Transwell inserts, which rely on diffusion-limited exchange and lack dynamic medium circulation, the POCC platform enables continuous biochemical communication under controlled flow conditions, thereby improving the physiological relevance of inter-tissue metabolic interactions.”

6. The 1:1 ratio of 3T3-L1: HepG2 cells should be justified.

Response: We thank the reviewer for this important comment. Defining a physiologically exact hepatic-to-adipose cell ratio that can be directly translated into an in vitro co-culture system is challenging, as adipose tissue mass, distribution, and metabolic activity are highly dynamic and vary with nutritional status, body composition, and disease conditions. Consequently, in liver–adipose co-culture models, the primary objective is not to replicate a fixed in vivo tissue proportion, but rather to enable controlled investigation of paracrine metabolic communication between compartments.

In the present study, a 1:1 seeding ratio of 3T3-L1 and HepG2 cells was selected based on preliminary optimization experiments and to provide balanced bidirectional signaling without dominance of either cell population. This ratio was chosen as a practical and reproducible starting point for this proof-of-concept platform and has now been discussed in the revised Discussion section (lines 437-449).

“Although physiological relevance is an important consideration in multi-tissue in vitro models, translating in vivo hepatic–adipose tissue proportions into a fixed in vitro cell ratio remains challenging [54]. Adipose tissue mass and metabolic activity are highly dynamic and vary substantially with nutritional status, body composition, and disease state, and different adipose depots contribute differently to systemic metabolism. These factors complicate the definition of a in vivo tissue ratio and highlight that simple mass-based ratios may not fully reflect physiological dynamics [55,56]. As a result, liver–adipose co-culture systems generally prioritize capturing paracrine metabolic communication rather than replicating exact in vivo tissue proportions. In this study, a 1:1 ratio of 3T3-L1 to HepG2 cells was therefore selected as a balanced and reproducible starting point to support bidirectional metabolic signaling between compartments, while maintaining experimental simplicity and robustness in this proof-of-concept platform. In Future studies, other ratios of hepatic to adipose cells could be investigated according to the scope of a study.”

7. Before positioning the system as a disease model or an alternative to animal testing, the hepatic compartment should be validated for core liver metabolic/synthetic function under perfusion, even at a phenotypic/directional level. At minimum, I recommend assessing:

• Albumin secretion (e.g., in device vs static control in 96-well plate)

• Urea production trends

Response: We thank the reviewer for this valuable suggestion. We agree that direct assessment of hepatic synthetic function, including albumin secretion and urea production, is essential for full validation of the hepatic compartment, particularly when positioning the system as a disease-relevant or animal-alternative model.

The present study was designed as a proof-of-concept platform to investigate liver–adipose metabolic interactions and drug-induced metabolic alterations under passive perfusion, rather than to establish a fully validated liver disease model. Accordingly, direct measurements of albumin and urea production were not included and are now explicitly acknowledged as a limitation of the current work.

This limitation and the incorporation of established hepatic functional markers, such as albumin and urea, as part of future validation studies have been clearly stated in the revised Discussion section (lines 515-526): “Albumin secretion and urea synthesis are widely accepted liver-specific functional markers for validating hepatic synthetic activity in vitro liver models and under dynamic and perfused culture conditions in liver-on-chip systems [78]. Such measurements are commonly incorporated in later-stage validation studies once platform feasibility, stability, and metabolic responsiveness have been established.

In the present study, the primary objective was not to establish a fully validated liver disease model, but rather to develop and apply a pumpless co-culture platform to investigate inter-tissue metabolic communication and drug-induced metabolic alterations between liver and adipose compartments. Accordingly, direct functional validation assays of hepatic synthetic activity, such as albumin and urea production, were not included and are acknowledged as a limitation of the current work. Future studies will incorporate these established functional endpoints to further validate hepatic performance within the POCC system, as demonstrated in prior perfused HepG2-based microfluidic models [79].”

8. The authors should expand discussion on why passive-flow/perfused culture better mimics in vivo compared to static systems, ideally correlating the device phenotype with

---

## [Editor Report · Decision Letter 1]

9 Mar 2026

A pumpless liver-adipose model for studying metabolic dysfunction and drug responses

PONE-D-25-60517R1

Dear Dr. Mousavi Shaegh,

We’re pleased to inform you that your manuscript has been judged scientifically suitable for publication and will be formally accepted for publication once it meets all outstanding technical requirements.

An invoice will be generated when your article is formally accepted. Please note, if your institution has a publishing partnership with PLOS and your article meets the relevant criteria, all or part of your publication costs will be covered. Please make sure your user information is up-to-date by logging into Editorial Manager at Editorial Manager® and clicking the ‘Update My Information' link at the top of the page. For questions related to billing, please contact  and clicking the ‘Update My Information' link at the top of the page. For questions related to billing, please contact billing support..

Kind regards,

Santhi Silambanan, MD, DNB

Academic Editor

PLOS One

---

## [Editor Report · Acceptance letter]

PONE-D-25-60517R1

PLOS One

Dear Dr. Mousavi Shaegh,

I'm pleased to inform you that your manuscript has been deemed suitable for publication in PLOS One. Congratulations! Your manuscript is now being handed over to our production team.

Kind regards,

on behalf of

Dr. Santhi Silambanan

Academic Editor

PLOS One